# Nepalese version of Douleur Neuropathique 4 (DN4) questionnaire for detection of neuropathic pain signs and symptoms: Translation and psychometric properties

Bigen Man Shakya[1] , Anil Shrestha[1], Amod Kumar Poudyal[2], Ninadini Shrestha[1], Binita Acharya[1], Renu Gurung[1], Sujata Shakya[2] *

1 Department of Anaesthesiology, Maharajgunj Medical Campus, Tribhuvan University Institute of Medicine, Maharajgunj, Kathmandu, Nepal, 2 Central Department of Public Health, Tribhuvan University Institute of Medicine, Maharajgunj, Kathmandu, Nepal

These authors contributed equally to this work.

* sujata_8@iom.edu.np

## Abstract

### Objective

This study aimed to translate the DN4 questionnaire into Nepalese version and assess its psychometric properties: diagnostic accuracy, internal consistency, and test-retest reliability.

### Methods

An observational study was conducted in a tertiary level teaching hospital of Kathmandu, Nepal. We included 166 patients with chronic pain visiting a pain clinic over a period of one year. The Nepalese version of the DN4 questionnaire was used for detecting signs and symptoms of neuropathic pain. The English version of the questionnaire was translated into Nepali in accordance with the standard guideline with the help of linguistic experts. The patients who met the inclusion criteria were examined and interviewed twice in an interval of two weeks. The association between the index test and the reference test was analyzed using Chi-square test. Diagnostic accuracy was assessed using sensitivity, specificity, Youden's index, and positive and negative predictive values. We calculated internal consistency using Cronbach's alpha ($\propto$), and test-retest reliability using Cohen's kappa and Intra-class correlation coefficient (ICC).

### Results

The study showed a significant association between the result of DN4 questionnaire and the gold standard (physician's diagnosis) ($p<0.001$). The sensitivity and specificity values for the DN4 questionnaire were 75% and 95.3% respectively. Similarly, positive and negative predictive values were 93.8% and 80.4% respectively. Our study showed adequate internal consistency ($\propto = 0.710$) and a good test-retest reliability (kappa = 0.872, ICC = 0.877).

**Data Availability Statement:** All relevant data are within the paper and its Supporting Information files.

**Funding:** The authors received no specific funding for this work.

**Competing interests:** The authors have declared that no competing interests exist.

## Conclusions

The Nepalese version of DN4 questionnaire is a valid and reliable tool for the identification of signs and symptoms of neuropathic pain. This can be used for screening neuropathic pain signs and symptoms in clinical as well as research settings.

## Introduction

Neuropathic pain is defined as pain caused by a lesion or disease of the somatosensory nervous system [1]. The diagnosis of neuropathic pain is complex, and it is based on clinical history, physical examination, and other advanced investigations such as neuro-physiological techniques (e.g. laser evoked potentials, quantitative sensory testing, skin biopsy) [2]. Neuropathic pain is one of the prevalent health conditions, with the prevalence of 7–10% [3]. Several questionnaires have been developed to facilitate the detection of neuropathic signs and symptoms. They include Neuropathic pain Scale [4], Pain detect [5], Leeds Assessment of Neuropathic Symptoms and Signs (LANSS) [6], Douleur Neuropathique 4 (DN4) etc. These screening tools consist of structured questions and some of them may include a simple clinical examination. The identification of neuropathic signs and symptoms can be done with a high degree of sensitivity and specificity with the help of aforementioned tools [7]. Moreover, these tools can be used by a specialist as well as any health personnel [8].

Due to its popularity, the DN4 questionnaire has been translated and validated into various languages such as Hindi [9], Thai [10], Japanese [11], Korean [12], Spanish [13], Greek [14], Portuguese [15], Turkish [16] and many more. It was originally developed by French Neuropathic pain group in 2005 in French and was later translated into English by the same team [17]. This screening tool is simple to use; we can incorporate it into daily clinical practices as well research studies.

Translation of the standard questionnaire should preserve the meaning and intent of the original item, and its validity and reliability must be maintained. The process of translation and its adaptation to different languages is demanding and time consuming. Sometimes inadequate translation could lead to misleading conclusions in clinical practice and/or research studies. In order to avoid this, there exist certain guidelines for the translation and validation of tools [18]. Some of the standard tools, like Numerical Pain Rating Scale, have already been translated into Nepali language and validated [19]. A scoping review by Sharma et. al showed that majority of the studies in pain in Nepal are published in local journals, and they mainly focus on the diagnosis and management of clinical pain [20]. There is a lack of studies providing information on basic statistics like prevalence of chronic pain conditions including neuropathic pain. The assessment of psychometric properties of internationally accepted tools is needed in Nepali so that they can be used uniformly for clinical and research purposes. The S-LANSS scale which was validated in 2021 is a subjective based questionnaire which helps to identify pain of predominantly neuropathic origin [21]. There is another standard screening tool for neuropathic pain (DN4 questionnaire) which is based on both interview and clinical examination, but it has not yet been interpreted in Nepali. Therefore, our aim is to translate this questionnaire into Nepalese language and assess its psychometric properties: diagnostic accuracy, internal consistency, and test-retest reliability.

## Materials and methods

We used checklist based on Standards for Reporting of Diagnostic Accuracy Studies (STARD) guidelines [22] and COnsensus-based Standards for the selection of health Measurement INstruments (COSMIN) study design checklist [23] (S1 Fig) for conducting and reporting our study.

### Study design

We conducted a prospective observational study among patients with chronic pain attending the pain clinic of Tribhuvan University Teaching Hospital, a tertiary level teaching hospital of Kathmandu, Nepal, over a period of one year from February 2019 to January 2020.

### Participants selection

We adopted nonprobability purposive sampling technique for the selection of patients and included adult patients (18 years and above) who attended the pain clinic with chronic pain (pain duration of more than 3 months) and patients referred from other departments for further evaluation of chronic pain. We recruited patients fluent in Nepali and with musculoskeletal pain involving shoulder, knee and wrist, neck and back with or without radiculopathy, trigeminal neuralgia, post-traumatic trigeminal neuropathy and post-herpetic neuralgia. Patients with fibromyalgia, phantom pain, headache, chronic visceral pain, cancer pain, and severe depression were excluded. This was done in accordance with the inclusion and exclusion criteria used for the validation of DN4 questionnaire in other language [24].

Due to the lack of consensus regarding the calculation of the sample size in validation studies, the suggested ratio of subject to item ranges from 2–20 [25]. The total sample size for this study was 166, which put the subject-to-item ratio at 16:1.

The ethical approval for the research was obtained from the Institutional Review Committee of Institute of Medicine, Tribhuvan University [338(6–11)E$^2$/075/76]. Written informed consent was taken from all patients before the data collection.

### Data collection

We collected the following demographic information: age in years, sex (male, female) and ethnicity (Brahmin/Chhetri, Janajati, Others) (S4 Fig). We used the Nepalese version of the questionnaire to interview chronic pain patients for the detection of the neuropathic pain signs and symptoms.

The DN4 questionnaire consists of 10 items, in which 7 items are related to pain characteristics and 3 items are related to findings from physical examination of the painful areas. The cut-off value for the diagnosis of neuropathic pain is 4/10. The area under the curve for the total score of French version of the questionnaire is 0.92. A cut-off score of 4 resulted in the highest percent of sensitivity (82.9%) and specificity (89.9%). The inter-rater reliability (Cohen Kappa coefficient) is between 0.70 and 0.96 [17].

**Phase 1: Translation and back-translation.** We adopted guidelines given by Sperber AD for the translation process [18] and also obtained permission from the principal author of the first DN4 questionnaire. The English version of the questionnaire was translated into Nepalese version independently by three individuals: two linguistic experts, and one neurologist who had experience in management of chronic pain. These three individuals along with the principal investigator then discussed discrepancies in their translations and formed a preliminary draft. The draft so formed was back translated into English by another translation expert who was not part of aforementioned team. The committee of three pain specialists (anaesthesiologists trained in management of chronic pain), along with the forward and backward

translators and a research expert discussed and reviewed the items of all versions of the translation and compared them with the original English version for semantic, experiential and conceptual equivalence. Finally, the penultimate Nepalese version of DN4 was created.

**Phase 2: Pretesting and modification.** A panel of experts containing a methodologist, a statistician and two pain specialists established the content validity of the questionnaire. The translated Nepalese version was then compared with the English version and discussed among the experts. We analyzed the interpretability of words, phrases, and sentences and examined the questionnaire to find out whether the items adequately measured the intended constructs and were sufficient to measure different domains or not. Afterwards, a translation revision was conducted based on consensus.

Pretesting of the penultimate Nepalese version was carried out among 15% of the total sample, which included 25 patients with neuropathic pain attending a private pain clinic situated in Kathmandu, Nepal. During the pretesting, the patients were specifically asked "Do you understand about items listed in the questionnaire?".

Face validity of the questionnaire was maintained by pretesting the pre-final version of the translated Nepalese DN4 questionnaire among 25 respondents. We found that all of the pain descriptors were well understood by the respondents except for the term 'painful cold'. Various words were explored in order to find the closest equivalence to the term 'painful cold'. Finally, the phrase "धेरै चिसो हुँदा खेरी हुने किसिमको दुखाई" (type of pain which occurs during extreme cold) was chosen as the most appropriate way to refer to it. The newly translated phrase was retested among 10 patients, which yielded the satisfactory results. Taking all this into account, the final version of the questionnaire was developed (S5 Fig).

## Tests

We assessed neuropathic pain signs and symptoms by comparing the index test (using DN4's dichotomous categories) and the reference standard (physician's diagnosis of pain). We divided the patients with chronic pain into two groups (neuropathic and non-neuropathic) after examination by a pain specialist at a pain clinic. The detection of neuropathic signs and symptoms was made through the patient's history, clinical examination (probable neuropathic pain) based on the guidelines provided by International Association for the Study of Pain (IASP) which is the internationally accepted guideline [26]. The decision of the pain physician was considered final in the probable diagnosis of neuropathic pain. This was considered as the reference test or gold standard. A pain specialist, who was not involved in the segregation of patients, then performed interviews on the same day using the final corrected Nepali version of DN4 questionnaire. The result were established as the index test. Those patients were interviewed again using the same questionnaire during follow-up within two weeks. We considered the cut-off value for leveling neuropathic pain by DN4 questionnaire as 4/10 and above. This cut-off was also originally used in the French version [17].

## Data analysis

We entered the data into Microsoft Excel 2013 and then exported to SPSS version 16 for statistical analysis. Any missing or indeterminate test results were excluded from the study. Under descriptive statistics, age was reported as mean and standard deviation, and sex and ethnicity were presented as frequency and percentage. Different items of DN4 questionnaire compared between neuropathic and non-neuropathic patients were reported as frequency and percentages.

**Diagnostic accuracy.** To compare the diagnosis of neuropathic pain signs and symptoms by the index test and the reference test, chi-square test was used. We dichotomized DN4 scores

into two categories ('presence of neuropathic pain signs and symptoms' and 'absence of neuropathic pain signs and symptoms') based on the cut-off score 4. Different diagnostic values to detect neuropathic pain signs and symptoms like sensitivity, specificity, Youden's index, and positive and negative predictive values were calculated. Sensitivity is the test's probability of correctly detecting condition among those with the condition, and this was calculated by number of positive tests based on DN4 questionnaire out of those with neuropathic pain signs and symptoms as diagnosed by physician. Specificity is the test's probability of giving negative result for those not having the condition, and we calculated this by number of negative tests based on DN4 questionnaire out of those with non-neuropathic pain signs and symptoms by physician's diagnosis [27]. Youden's index measures a diagnostic test's ability to balance sensitivity and specificity [28]. This was calculated by adding the sensitivity of the result of the DN4 questionnaire to the specificity of the same result, and then subtracting 1 from that value. Similarly, Positive Predictive Value (PPV) refers to the likelihood of the person having the condition with the test positive, whereas, Negative Predictive Value (NPV) refers to the likelihood of the person being healthy when the test is negative [29]. PPV was calculated by number of positive tests based on DN4 questionnaire among those diagnosed with neuropathic pain signs and symptoms by the physician, while NPV was calculated by number of negative tests based on DN4 questionnaire among those diagnosed with non-neuropathic pain signs and symptoms by the physician. Higher the percentage of these indicators, the more accurate the test is in diagnosing the condition.

**Internal Consistency.** To assess the internal consistency, Cronbach's alpha ($\alpha$) was calculated in which $\alpha \geq 0.7$ was considered adequate or acceptable [30].

**Test-retest reliability.** The test-retest reliability was calculated using Cohen's kappa statistics for dichotomous categories, and ICC for DN4 scoring 0–10. Kappa$\leq$0 indicates no agreement, 0.01–0.20 none to slight, 0.21–0.40 fair, 0.41–0.60 moderate, 0.61–0.80 substantial, and 0.81–1.00 almost perfect agreement [31]. For ICC, we used two-way random effects, absolute agreement, single rater measurement, that is ICC (2,1) [32,33]. ICC values <0.5 indicates poor reliability, 0.5–0.75 moderate reliability, 0.75–0.9 good reliability, and >0.90 excellent reliability [33].

## Results

### Demographic characteristics

The total participants attending the pain clinic during the study period was 216. Among them, 50 patients were excluded due to various reasons like cancer, headache, pain less than 3 months, phantom pain, and visceral pain. Thus, eligible 166 patients were enrolled which is presented in the flow diagram (S3 Fig). There was no missing data in our study. First time we encountered a problem while we were trying to restate questions 1 and 2 in Nepali. [Question 1: Does the pain have one or more of the following characteristics? Question 2: Is the pain associated with one or more of the following symptoms in the same area?] Due to the inherent discrepancies in the grammatical rules and lexicons of the two languages, achieving a direct rendering of the aforementioned questions proved infeasible. With the aid of two interpreters and clinical staff, we managed to come up with restatements that are close in meaning to the original questions, and also in line with the conventions of the Nepali language. All items in the questionnaire have Nepali equivalents except for 'painful cold'. We rephrased this difficult word that conveyed the same meaning as 'painful cold'.

The general characteristics of the respondents are presented in Table 1. The mean age of the patients with neuropathic and non-neuropathic pain signs and symptoms were 49.03±15.1 and 52.79±14.8 years respectively. Majority were females and Brahmin/Chhetri by ethnicity.

**Table 1. Baseline characteristics of the respondents.**

| Characteristics | | Non-neuropathic (n = 86) | Neuropathic (n = 80) |
|---|---|---|---|
| **Age (in years)** | Mean age | 49.03±15.1 | 52.79±14.8 |
| **Sex** | Male | 28 (32.6%) | 31 (38.8%) |
| | Female | 58 (67.4%) | 49 (61.3%) |
| **Ethnicity** | Brahmin/Chhetri | 45 (52.3%) | 39 (48.8%) |
| | Janajati | 32 (37.2%) | 31 (38.8%) |
| | Others* | 9 (10.5%) | 10 (12.5%) |

*Others include Dalit, Madhesi and Muslim.

## Comparison between neuropathic and non-neuropathic pain signs and symptoms

The most common symptoms in patients with neuropathic pain were tingling sensation (75.0%), burning pain (72.5%), pins and needles (71.3%), and electrical shock (71.3%) (Table 2).

Table 3 compares the diagnosis of neuropathic pain signs and symptoms by the index test and the physician's gold standard. It shows significant association between the index test and reference test (p<0.001).

Table 4 illustrates different measures of diagnostic accuracy of the test, like sensitivity, specificity, Youden's index, and positive and negative predictive values.

## Internal consistency (Cronbach's alpha)

The Cronbach's alpha ($\propto$) coefficient of the entire DN4 questionnaire is found to be 0.710, indicating adequate internal consistency. The coefficient value was not much different by removing any of the items in the questionnaire. Thus, all questions were included in the DN4 questionnaire (Table 5).

## Test–retest reliability

The patients were interviewed twice using the Nepalese version of DN4 questionnaire, once during their first visit at the pain clinic, and again within 2 weeks. The test-retest reliability was calculated using Cohen's kappa statistics (Table 6), and ICC (S2 Fig). The Kappa coefficient

**Table 2. Frequency of pain signs and symptoms based on DN4 questionnaire.**

| Items | Non-neuropathic (n = 86) | Neuropathic (n = 80) |
|---|---|---|
| | n (%) | n (%) |
| **Tingling** | 20 (23.3) | 60 (75.0) |
| **Burning** | 25 (29.1) | 58 (72.5) |
| **Electric shock** | 12 (14.0) | 57 (71.3) |
| **Pins and needles** | 30 (34.9) | 57 (71.3) |
| **Numbness** | 5 (5.8) | 35 (43.8) |
| **Hypoesthesia to touch** | 2 (2.3) | 27 (33.8) |
| **Increased pain on brushing** | 3 (3.5) | 25 (31.3) |
| **Hypoesthesia to pin prick** | 0 (0.0) | 24 (30.0) |
| **Painful cold** | 7 (8.1) | 14 (17.5) |
| **Itching** | 1 (1.2) | 14 (17.5) |

**Table 3. Comparison of the findings of DN4 questionnaire with physician's diagnosis of pain.**

| Pain assessment using DN4 questionnaire | Reference test (Physician's diagnosis) | |
| --- | --- | --- |
| | Non-neuropathic (n = 86) | Neuropathic (n = 80) |
| | n (%) | n (%) |
| Negative | 82 (80.4) | 20 (19.6) |
| Positive | 4 (6.3) | 60 (93.8) |

was 0.872, showing perfect agreement between the test at baseline and the test at follow-up. Similarly, the ICC value was found to be more than 0.8, indicating a good test-retest reliability value.

## Discussion

This is the first study to translate and validate DN4 questionnaire into Nepali. The Nepalese version of DN4 tool demonstrated acceptable diagnostic accuracy and reliability. The index test and the reference test showed significant association with each other.

### Translation and cultural adaptation

During our clinical practice, the patients used different terminologies to describe the symptoms of electric shock, for instance, '*silka hanne*', '*shola hanne*', '*jhatka haneko*', '*current lageko*'. The translators who translated it from English to Nepali had initially chosen phrases like '*current lageko jasto*' and '*bidutiya jhatka*'. Although '*current*' is not a Nepali word, it is easily understood by many people. Therefore, we decided on '*current lageko* jasto' in our final draft. Most of the patients actually thought that painful cold means pain aggravated by cold weather. For this reason, caution must be exercised while asking the patient about painful cold.

### Response of the questionnaire

Based on our study, the most frequent complaints that the patients with neuropathic pain had were burning pain, electrical shock, tingling sensation, and numbness. When this is compared with other studies, we found variations [9,34,35]. We believe that neuropathic pain consists of a collection of symptoms and signs that may also depend on the type of disease, so rather than focusing on any specific item for predicting neuropathic pain, one must consider the cut-off value.

### Diagnostic accuracy

The sensitivity and specificity of our version of the questionnaire were 75% and 95.3%, respectively. In line with our findings, the original French version of the DN4 questionnaire has sensitivity of 82.9% and specificity of 89.9% [17]. The Japanese version of the questionnaire has sensitivity and specificity of 71% and 92% respectively, which is almost similar to our findings [11]. However, the Spanish version has comparatively lower sensitivity (79.8%) and specificity

**Table 4. Diagnostic values of the DN4 questionnaire for the discrimination of neuropathic pain signs and symptoms.**

| Test Result | Sensitivity (%) | Specificity (%) | Youden's index | PPV (%) | NPV (%) |
| --- | --- | --- | --- | --- | --- |
| DN4 questionnaire | 75.0 | 95.3 | 0.7 | 93.8 | 80.4 |

PPV: Positive Predictive Value, NPV: Negative Predictive Value.

**Table 5. Internal consistency.**

| Item | Cronbach's alpha | Cronbach's alpha (if single item deleted) |
|---|---|---|
| Painful cold | 0.710 | 0.728 |
| Increased pain on brushing | | 0.719 |
| Pins and needles | | 0.710 |
| Itching | | 0.701 |
| Burning | | 0.689 |
| Electric shock | | 0.677 |
| Numbness | | 0.664 |
| Hypoesthesia to pin prick | | 0.661 |
| Hypoesthesia to touch | | 0.660 |
| Tingling | | 0.656 |

(78%) [13], which might be due to inclusion of mixed pain. This was also demonstrated by a study by Timmerman et.al [34]. There is, however, validation done in New Arabic language in which the cut-off value was taken as 5/10 [35]. The New Arabic version reported higher sensitivity (93%) compared to our findings, which might be attributed to different cut-off values.

## Reliability

While interviewing twice using the Nepali version of the questionnaire, it showed a good test-retest reliability value (kappa = 0.872, ICC = 0.8). In the same way, the New Arabic version of the questionnaire also showed excellent test-retest reliability (ICC>0.9) [35]. The test–retest intra-class correlation coefficient (95% CI) of the Japanese version was 0.827 (0.769–0.870) [11].

## Strengths and limitations

This study has assessed the psychometric properties of Nepalese version of DN4 questionnaire. The patients were first seen by a pain physician who was not involved in the interview. This maintains blinding that ensures the robustness of the diagnostic accuracy. However, there were few limitations too. The diagnosis of neuropathic pain signs and symptoms (reference test) is based only on history taking and clinical examination. Other investigations such as nerve conduction test, skin biopsy etc. could not be carried out due to financial constraints of the patients. Importantly, the institutional policy did not permit such investigations just for the sake of research. The same clinical diagnosis was used as reference in our study. Besides, Nepal is a small country, but with multicultural and multilinguistic population. So, this questionnaire is not applicable to those who cannot effectively communicate in Nepali. We did not assess education of the patients; therefore the socio-economic factors that could have affected the results could not be identified. Lastly, we could not assess construct validity by comparing with other Nepalese tools as none were available at the time of our study.

**Table 6. Test-retest reliability of the pain assessment at baseline and at follow-up using DN4 questionnaire.**

| Pain assessment at baseline | Pain assessment at follow-up | | | Kappa | p value |
|---|---|---|---|---|---|
| | Negative n (%) | Positive n % | Total | | |
| Negative | 98 (96.1) | 4 (3.9) | 102 (100.0) | 0.872 | 0.039 |
| Positive | 6 (9.4) | 58 (90.6) | 64 (100.0) | | |
| Total | 104 (62.7) | 62 (37.3) | | | |

### Implications to clinical practice and research

This study will provide opportunity for the clinical researchers in Nepal to use the standard tool in Nepali for detecting neuropathic pain signs and symptoms. This will help to promote clinical utility of the DN4 questionnaire. The DN4 questionnaire is based on interview and patient examination. Currently, there is validated subjective questionnaire in Nepali (S-LANSS), which has been tested on people with low literacy [21]. Now, the researchers will have a choice of tools which can be used as per their feasibility.

## Conclusion

The Nepalese version of DN4 questionnaire can now be used as a screening tool for identifying neuropathic pain signs and symptoms in patients with chronic pain. Further study is recommended in future for comparing two different Nepalese version tools (DN4 and SLANSS).

## Supporting information

**S1 Fig. COSMIN Checklist.**
(TIF)

**S2 Fig. Test-retest reliability of the Nepali version of the DN4 questionnaire using ICC.**
(TIF)

**S3 Fig. Participant flow diagram.**
(TIF)

**S4 Fig. Demographic information.**
(TIF)

**S5 Fig. Study questionnaire in Nepalese version.**
(TIF)

**S1 File. Validation study datasheet.**
(SAV)

## Acknowledgments

We would like to thank Late Mr. Laxman Rajbanshi (MA in Political Science), Mr. Tirtha Ratna Shakya (MA) and Dr. Sarbottam Shrestha (Neurologist) for translating the questionnaire into Nepali, and Mr. Sandesh Shakya (MA in English) for back translation. We would also like to thank the hospital and the patients who participated in the study.

## Author Contributions

**Conceptualization:** Bigen Man Shakya.

**Data curation:** Bigen Man Shakya, Sujata Shakya.

**Formal analysis:** Amod Kumar Poudyal, Sujata Shakya.

**Investigation:** Bigen Man Shakya, Ninadini Shrestha, Binita Acharya, Renu Gurung.

**Methodology:** Sujata Shakya.

**Project administration:** Bigen Man Shakya.

**Resources:** Bigen Man Shakya, Anil Shrestha, Ninadini Shrestha, Binita Acharya, Renu Gurung.

**Software:** Sujata Shakya.

**Supervision:** Bigen Man Shakya.

**Validation:** Bigen Man Shakya, Sujata Shakya.

**Writing – original draft:** Bigen Man Shakya, Sujata Shakya.

**Writing – review & editing:** Bigen Man Shakya, Anil Shrestha, Amod Kumar Poudyal, Ninadini Shrestha, Binita Acharya, Renu Gurung, Sujata Shakya.

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
