## [Decision Letter · Decision Letter 0]

11 Aug 2022

PONE-D-21-36866Nepalese version of Douleur Neuropathique 4 ( DN4) questionnaire for assessment of neuropathic pain: a validation studyPLOS ONE

Dear Dr. Shakya,

Thank you for submitting your manuscript to PLOS ONE. After careful consideration, we feel that it has merit but does not fully meet PLOS ONE’s publication criteria as it currently stands. Therefore, we invite you to submit a revised version of the manuscript that addresses the points raised during the review process.

Please see the comments from the reviewers below. Although both reviewers request that you add more introductory material on neuropathic pain, it is not essential that you cite the papers on foot pain noted by the first reviewer.

We look forward to receiving your revised manuscript.

Kind regards,

Steve Zimmerman, PhD

Associate Editor, PLOS ONE

Journal Requirements:

Reviewers' comments:

Reviewer's Responses to Questions

**Comments to the Author**

1. Is the manuscript technically sound, and do the data support the conclusions?

Reviewer #1: Partly

Reviewer #2: Partly

2. Has the statistical analysis been performed appropriately and rigorously? 

Reviewer #1: Yes

Reviewer #2: Yes

3. Have the authors made all data underlying the findings in their manuscript fully available?

Reviewer #1: Yes

Reviewer #2: No

4. Is the manuscript presented in an intelligible fashion and written in standard English?

Reviewer #1: Yes

Reviewer #2: No

5. Review Comments to the Author

Reviewer #1: I am grateful for the possibility to revise this research study.

The assessment of neuropathic pain is a trend topic in the current research literature and may be a main focus of interest for readers.

This is a well-written manuscript with an important clinical message, and should be of great interest to the journal readers. However, from my point of view, authors should include the following requeriments

Introduction may be improved adding new information in order to provide an adequate state-of-the-art including some references. I suggest to include this references include in the attached to complete this requirement related to neuropathic Diabetic foot complications that authors do not included

DOI 10.1111/iwj.13263

Methods are well-designed with relevant and complete information. Correct sample size calculations, good description of the properties of the outcome measurements as well as detailed statistical analyses were included.

Discussion section is well structured with different sections. Authors manage well the discussion leading a good comparison with the showed references.

However, author should discuss their results with regards other prior studies in validation and transcultural adaptation I suggest to include these references to attend this requeriment

doi:10.3390/ijerph17176141

DOI: 10.18632/aging.202140

Reviewer #2: Thank you for the opportunity to review this interesting manuscript. Translation and cross-cultural adaptation of outcome measures are necessary and critical to advance research and clinical care. I congratulate authors on leading this for neuropathic pain. In general, paper is fairly written and meets a minimum criteria for an academic paper. However, several suggestions are provided below in order to improve the story on why this study is important, how to improve reporting of the Methods, Results and Discussion.

Major criticisms:

1. The Introduction does not exactly cover why the study is needed. I would suggest the authors to include what is known about the topic in relation to Nepal, what the current gap is and how this paper addresses this major literature gap. You will see specific suggestions below.

2. Methods: Critical steps and items of validation studies are not covered. I would recommend authors to use COSMIN Checklist to report the current paper. https://www.cosmin.nl/wp-content/uploads/COSMIN-study-designing-checklist_final.pdf

3. Results: Also use COSMIN checklist to report results.

4. Discussion: It is quite vague and non-focused. I would recommend authors to rewrite the Discussion section to keep it more focused. Please focus on what key results were, whey they mean, how are the similar/different from other studies in the same country/region (or international literature if local/regional research is lacking), what are the implications, limitations, recommendations for the future studies. To write more focused Discussion section, authors may refer to a BMJ editorial, https://www.bmj.com/content/318/7193/1224

5. The authors state at the end of introduction "Once validated, this tool can be used as a screening tool by any health personnel for the diagnosis of neuropathic pain." In order to facilitate this more widely, I strongly recommend authors to upload Nepali DN4 as either appendix or supplementary online file to promote wide use of the scale in clinical practice or research. This aligns quite closely with ethos of PLOS ONE. I also recommend authors to share their data openly.

6. The authors indicate that DN4 is used to "diagnose neuropathic pain". Rather it is to identify signs and symptoms related to neuropathic pain. This is an important limitation of a self-reported measure and I request authors to change this consistently throughout the manuscript.

Specific comments:

INTRODUCTION:

1. Information presented in the Introduction is quite generic. I think authors should be more specific. Rather than only saying neuropathic pain is one of the prevalent conditions, also say what the prevalence is (Line 72); be specific on what other advanced investigations are and with citation please (line 71).

2. As above, Line 76: Rather than detection of neuropathic pain, it is the identification of the "neuropathic signs and symptoms".

3. When authors indicated the previous translation of pain related instruments in Nepali, I was hoping to see their use in the current study to assess construct validity of the scale, but this was lacking. It is the strength that authors have mentioned prior translation of numerical pain rating scale in Nepali. But what is missing is "what is known about Neuropathic pain" in Nepal and availability of other screening tools in Nepali. On a quick PubMed search, self-reported LANSS has been translated and validated in Nepali (https://pubmed.ncbi.nlm.nih.gov/34583020/). The Introduction would benefit by such review of literature and to highlight any advantages of availability of DN-4 in addition to self-reported LANSS and other instruments.

METHODS:

4. could the authors add the name of the tertiary care hospital.

5. Line 105: I would suggest to use simple term such as "Nepali" than to refer to the language by "Nepalese language".

6. Line 106: Authors have provided comprehensive list of conditions which were excluded. It is unclear why these were excluded. Additionally, it is also important to know what are the types of chronic pain conditions that were included.

7. Line 123: Info about translation guideline belongs to the next paragraph on Phase 1: Translation and back translation.

8. It is surprising to see 6 forward translations of the scale, I am unclear of the rationale why. It seems like rather waste of Human Resources. The paper authors cited does seem to support this.

9: Please provide some info on why 25 patients were recruited for pre-testing.

10. Line 148: State in how many people was the translation of "cold pain" was tested.

11. Line 150: It was odd that content validity of the scale was discussed after pre-testing. Ideally it is performed before pre-testing. Only upon all experts agree on the prefinal version, it is subjected to pretesting.

12. Test methods: Please explicitly state what was the diagnostic criteria based on the IASP used in the current study. This is important for the internal validity of the study.

13. specify how test-retest reliability was assessed (agreement versus consistency)? Also specify Fleiss notation system (Shrout & Fleiss, 1979), which describes the type of ICC for example as ICC(2,1).

14. Line 178: Please describe the method how ROC curve was plotted and how AUC was derived. State what the scores of AUC mean.

15. Line 180: Explain what each of the parameters mean (sensitivity, specificity and liklihood ratios) with explanation on how these were computed. State what these scores mean.

RESULTS:

16: The result would benefit with a flow diagram. How many patients were approached, how many recruited (and excluded), how many completed baseline survey, how many followed up at two weeks.

17. What was the mean duration of the follow up?

18. Could the authors provide any info on other sociodemographic variables such as ethnicity, education, occupation etc. These are important sociodemographic parameters that relates with the external validity and utility of the scale.

19. Table 4. Unclear what the ICC means in this Table. The results on ICC following this (text and Table 5) is quite high compared to what is presented in Table 4. Is it supposed to be pin prick?

20. Table 6 is an intense table. It should be simiplied so that it is self explanatory.

DISCUSSION:

see the major criticism comments above.

21. Limitations: An important aspect of construct validity, that is hypothesis testing is missing. That is testing strength and direction of association of the scale with other established scales (that is translated, cross-culturally adapted and validated scales) that assess the same or related constructs. This could be a recommendation for the future studies.

22. Limitations: Unclear who the study findings can be generalised to, because the full demographics of the study participants are unclear (e.g., sociocultural and economic positioning including education, ethnicity, occupation, income levels)

CONCLUSIONS:

As above, rather than predicting "neuropathic pain" it should be "identifying neuropathic pain signs and symptoms".

6. PLOS authors have the option to publish the peer review history of their article (what does this mean?). If published, this will include your full peer review and any attached files.

Reviewer #1: No

Reviewer #2: No

---

## [Author Response · Author response to Decision Letter 0]

10 Sep 2022

POINT BY POINT RESPONSES TO THE COMMENTS OF THE EDITOR AND REVIEWERS

PONE-D-21-36866

Response to the reviewers 

• Response: We thank the reviewers for reviewing the manuscript and very useful and encouraging comments. We have revised the manuscript in line with your suggestions. The specific changes and responses to the different points raised are mentioned below. Your comments are highlighted in bold followed by our responses.

Response to reviewer 1

The assessment of neuropathic pain is a trend topic in the current research literature and may be a main focus of interest for readers.

This is a well-written manuscript with an important clinical message, and should be of great interest to the journal readers. However, from my point of view, authors should include the following requeriments

Introduction may be improved adding new information in order to provide an adequate state-of-the-art including some references. I suggest to include this references include in the attached to complete this requirement related to neuropathic Diabetic foot complications that authors do not included

DOI 10.1111/iwj.13263

• Response: Thank you for the valuable comments and suggestion. As suggested by the Associate Editor of the journal that it is not essential to cite the papers on foot pain, we have not included the mentioned references in the introduction. However, we have added few references as suggested by reviewer 2, which we have mentioned under the response to reviewer 2.

Methods are well-designed with relevant and complete information. Correct sample size calculations, good description of the properties of the outcome measurements as well as detailed statistical analyses were included.

Discussion section is well structured with different sections. Authors manage well the discussion leading a good comparison with the showed references.

However, author should discuss their results with regards other prior studies in validation and transcultural adaptation I suggest to include these references to attend this requeriment

doi:10.3390/ijerph17176141

DOI: 10.18632/aging.202140

• Response: Thanks for your suggestion. We went through the mentioned literature. The first one was observational study and the second one was on validation of Edmonton Frail Scale (EFS), which is used to predict frailty disability outcomes among elderly. This tool is entirely different from the tool that we studied, which was meant to identify the neuropathic signs and symptoms. Therefore, as suggested by the Associate Editor, we have not included the mentioned references to compare our findings.

Response to reviewer 2

Major criticisms:

1. The Introduction does not exactly cover why the study is needed. I would suggest the authors to include what is known about the topic in relation to Nepal, what the current gap is and how this paper addresses this major literature gap. You will see specific suggestions below.

• Response: We have revised the introduction section mentioning the need for the validation of the DN4 questionnaire and the current status about the literature on neuropathic pain, which is lacking in our country. We have included recently published review article by Sharma et. al (2019) about the state of clinical pain research in Nepal. 

2. Methods: Critical steps and items of validation studies are not covered. I would recommend authors to use COSMIN Checklist to report the current paper. https://www.cosmin.nl/wp-content/uploads/COSMIN-study-designing-checklist_final.pdf

• Response: We totally agree with the reviewer and have referred to the COSMIN checklist to report the methods.

3. Results: Also use COSMIN checklist to report results.

• Response: Thank you for the suggestion. We have checked the results with the COSMIN checklist. We have also attached the checklist in the separate file.

4. Discussion: It is quite vague and non-focused. I would recommend authors to rewrite the Discussion section to keep it more focused. Please focus on what key results were, whey they mean, how are the similar/different from other studies in the same country/region (or international literature if local/regional research is lacking), what are the implications, limitations, recommendations for the future studies. To write more focused Discussion section, authors may refer to a BMJ editorial, https://www.bmj.com/content/318/7193/1224

• Response: We have divided the discussion into three parts, the first involved the difficulties we faced during the translation process. In the second part, we compared our findings with those of other literature. We have added few more references for discussing and comparing our findings. Likewise, in the third part, we have stated the study limitations, implications for future studies and conclusion.

5. The authors state at the end of introduction "Once validated, this tool can be used as a screening tool by any health personnel for the diagnosis of neuropathic pain." In order to facilitate this more widely, I strongly recommend authors to upload Nepali DN4 as either appendix or supplementary online file to promote wide use of the scale in clinical practice or research. This aligns quite closely with ethos of PLOS ONE. I also recommend authors to share their data openly.

• Response: We have already uploaded Nepali DN4 tool as the supplementary file during online submission of our manuscript. You can check it under our submission portal.

6. The authors indicate that DN4 is used to "diagnose neuropathic pain". Rather it is to identify signs and symptoms related to neuropathic pain. This is an important limitation of a self-reported measure and I request authors to change this consistently throughout the manuscript.

• Response: We highly appreciate and agree with this comment by the reviewer. As suggested, we have revised accordingly throughout the manuscript.

Specific comments:

INTRODUCTION:

1. Information presented in the Introduction is quite generic. I think authors should be more specific. Rather than only saying neuropathic pain is one of the prevalent conditions, also say what the prevalence is (Line 72); be specific on what other advanced investigations are and with citation please (line 71).

• Response: We thank the reviewer for the comment. We have tried to make the introduction more specific by mentioning the exact prevalence of the neuropathic pain. We also listed what other advanced investigations are. 

2. As above, Line 76: Rather than detection of neuropathic pain, it is the identification of the "neuropathic signs and symptoms".

• Response: We totally agree with the reviewer. We have now changed the sentence as per the suggestion.

3. When authors indicated the previous translation of pain related instruments in Nepali, I was hoping to see their use in the current study to assess construct validity of the scale, but this was lacking. It is the strength that authors have mentioned prior translation of numerical pain rating scale in Nepali. But what is missing is "what is known about Neuropathic pain" in Nepal and availability of other screening tools in Nepali. On a quick PubMed search, self-reported LANSS has been translated and validated in Nepali (https://pubmed.ncbi.nlm.nih.gov/34583020/). The Introduction would benefit by such review of literature and to highlight any advantages of availability of DN-4 in addition to self-reported LANSS and other instruments.

• Response: Thank you for the comment. The validation of Nepali Version of S-LANSS for assessment of neuropathic signs and symptoms was published only in March, 2022 by Saurav Sharma et. al. This was after completion of our data collection. Therefore, we did not find any other validated Nepali tool to include in our research. 

METHODS:

4. Could the authors add the name of the tertiary care hospital.

• Response: Thanks a lot for the suggestion. We have now mentioned the name of the tertiary care hospital as ‘Tribhuvan University Teaching Hospital’.

5. Line 105: I would suggest to use simple term such as "Nepali" than to refer to the language by "Nepalese language".

• Response: As suggested by the reviewer, we changed the word ‘Nepalese language’ to ‘Nepali’.

6. Line 106: Authors have provided comprehensive list of conditions which were excluded. It is unclear why these were excluded. Additionally, it is also important to know what are the types of chronic pain conditions that were included.

• Response: We totally agree with the reviewer’s comment. We specifically mentioned what types of chronic pain conditions were included and what were the reasons for the exclusion.

7. Line 123: Info about translation guideline belongs to the next paragraph on Phase 1: Translation and back translation.

• Response: As suggested by the reviewer, we moved this line under phase 1.

8. It is surprising to see 6 forward translations of the scale, I am unclear of the rationale why. It seems like rather waste of Human Resources. The paper authors cited does seem to support this.

• Response: Thank you for the comments. I would like to make it clear that the forward translation of the scale was performed by only three individuals not 6, which is clearly mentioned in phase 1. Among the three, two were linguistic experts and one was neurologist.

9: Please provide some info on why 25 patients were recruited for pre-testing.

• Response: We have mentioned in Line 158 the reason for recruiting 25 patients for pre-testing.

10. Line 148: State in how many people was the translation of "cold pain" was tested.

• Response: Thank you for the comment. We have now clearly specified the total number of participants who were tested for cold pain.

11. Line 150: It was odd that content validity of the scale was discussed after pre-testing. Ideally it is performed before pre-testing. Only upon all experts agree on the prefinal version, it is subjected to pretesting.

• Response: We totally agree with the reviewer on this comment. The content validity of the scale was assessed before pre-testing, which was mistakenly mentioned after pre-testing in the earlier version of the manuscript. Now, it has be corrected.

12. Test methods: Please explicitly state what was the diagnostic criteria based on the IASP used in the current study. This is important for the internal validity of the study.

• Response: In the study limitation, we have mentioned that the diagnosis of neuropathic pain by the pain experts based on history taking and clinical examination was taken as standard, which is considered to be the probable neuropathic pain as per IASP guidelines. For the diagnosis of definitive neuropathic pain, advanced tests have to be used, which was not possible in our case due to financial constraints. So, we have revised our statement in the manuscript (Line 181-182)

13. specify how test-retest reliability was assessed (agreement versus consistency)? Also specify Fleiss notation system (Shrout & Fleiss, 1979), which describes the type of ICC for example as ICC(2,1).

• Response: We have mentioned both in the methods and results section that the test-retest reliability was assessed in two episodes in 2 weeks period. As per the Fleiss notation system, we used type C intraclass correlation coefficients, that is, ICC (3,1) which is a two-way mixed effects, consistency, single rater measurement. This was clearly explained in line 222-223.

14. Line 178: Please describe the method how ROC curve was plotted and how AUC was derived. State what the scores of AUC mean.

• Response: As per the suggestion of the reviewer, we have explicitly explained how ROC curve was plotted and how AUC was derived and interpreted in line 201-205.

15. Line 180: Explain what each of the parameters mean (sensitivity, specificity and liklihood ratios) with explanation on how these were computed. State what these scores mean.

• Response: All the parameters and their interpretations are explained in the methods section in line 209-217

RESULTS:

16: The result would benefit with a flow diagram. How many patients were approached, how many recruited (and excluded), how many completed baseline survey, how many followed up at two weeks.

• Response: We had shown the patient flow diagram as a supplementary file, where we presented the number of potentially eligible participants and the number recruited and excluded due to various reasons. We have now slightly changed the diagram and tried to simplify it as per the suggestion of the reviewer. 

17. What was the mean duration of the follow up?

• Response: The follow up of the patients was done within 2 weeks of the first visit. The patients visited the clinic as per their convenience and schedules of the pain physician. All the follow ups were either at the 7th day or 14th day depending on their symptoms. However, we did not calculate the mean duration of the follow up.

18. Could the authors provide any info on other sociodemographic variables such as ethnicity, education, occupation etc. These are important sociodemographic parameters that relates with the external validity and utility of the scale.

• Response: We have added the ethnicity of the participants in Table 1. We did not look into the other variables like education and occupation since we did not feel much relevance of these variables with neuropathic pain based on our clinical experience.

19. Table 4. Unclear what the ICC means in this Table. The results on ICC following this (text and Table 5) is quite high compared to what is presented in Table 4. Is it supposed to be pin prick?

• Response: It is mentioned in the footnote of table 4 what the ICC denotes. Table 4 shows the ICC of internal consistency reliability that measures how well the tool addresses different constructs. On the other hand, Table 5 shows the ICC of the test-retest reliability. Therefore, these values came different. Yes, we have corrected the word pin prick.

20. Table 6 is an intense table. It should be simiplied so that it is self explanatory.

• Response: Thank you for the comments. We have tried to make the table relatively simple. We removed the scores of positive and negative likelihood ratios as these were derived out of the sensitivity and specificity values which we have already showed in the table.

DISCUSSION:

see the major criticism comments above.

21. Limitations: An important aspect of construct validity, that is hypothesis testing is missing. That is testing strength and direction of association of the scale with other established scales (that is translated, cross-culturally adapted and validated scales) that assess the same or related constructs. This could be a recommendation for the future studies.

• Response: During our study period, there was no other validated Nepali scale for testing the neuropathic signs and symptoms. The LANSS scale which was validated by Sharma et. al in Nepali was published only in 2022. In future studies, both of these tools can be used for testing the construct validity.

22. Limitations: Unclear who the study findings can be generalised to, because the full demographics of the study participants are unclear (e.g., sociocultural and economic positioning including education, ethnicity, occupation, income levels)

• Response: Our study findings can be generalized to all those who can communicate in Nepali, irrespective of their demographics like education, ethnicity, occupation and income. We also mentioned in the discussion that we did not face any difficulty during data collection because of the difference in literacy level.

CONCLUSIONS:

As above, rather than predicting "neuropathic pain" it should be "identifying neuropathic pain signs and symptoms".

• Response: The word has been revised as per the suggestion.

---

## [Decision Letter · Decision Letter 1]

17 Nov 2022

PONE-D-21-36866R1Nepalese version of Douleur Neuropathique 4 ( DN4) questionnaire for assessment of neuropathic pain: a validation studyPLOS ONE

Dear Dr. Shakya,

Thank you for submitting your manuscript to PLOS ONE. After careful consideration, we feel that it has merit but does not fully meet PLOS ONE’s publication criteria as it currently stands. Therefore, we invite you to submit a revised version of the manuscript that addresses the points raised during the review process.

We look forward to receiving your revised manuscript.

Kind regards,

Saurab Sharma, Ph.D.

Guest Editor

PLOS ONE

Additional Editor Comments:

Dear Dr Shakya and colleagues,

Thank you for your patience related to the review process of this paper. I must acknowledge that I was one of the reviewers in the previous round of the review. The PLOS ONE editorial office requested me to serve as the associate editor for your paper. I accepted the request for two reasons: (1) I am familiar with this area of research very closely, both at the global and local context of Nepal, and (2) I wanted to help make an early decision as I know how frustrating this is to the authors, especially because the paper is under review for almost a year now. I should also mention that before accepting this request, I also declared to the editorial board that “I have published extensively in the area of pain in Nepal. The authors of this paper have not reviewed literature well and cited previous work appropriately. This bias will remain in me when making the decision.”

The editorial decision is based on the assessment of your paper by the previous editor and reviewer and a new reviewer, in addition to my own assessment. In addition to the new reviewer’s feedback, please address my own concerns below. Please assist me in making an early decision, by responding to the reviewer’s and editor's comments fully. You are free to refuse any of the suggestions but fully justify this using credible references. Please write your responses in full, cite line numbers where changes were made, and also in your response letter, indicate what changes were made. That is, copy and paste key changes made in the manuscript into your response letter in order to assist the editor and reviewers.

1. The Introduction does not completely cover “what is known” and the need for the study or “what this study adds to what is known” at the time of writing the paper. Consistent with the comments in the previous round of review, the Introduction needs to cover relevant literature on (neuropathic) pain in Nepal. I appreciate you cited Sharma et al (2019) Pain Reports paper to identify gaps in the neuropathic literature. However, a paper published online in September 2021 (Journal of Pain; https://pubmed.ncbi.nlm.nih.gov/34583020/ ), identified the prevalence of neuropathic pain signs and symptoms as 12% in the general population and 23% in people with chronic pain. This is probably the only relevant publication on the topic in Nepal so far. The authors also indicated that the paper was published after they conceived the study, which is a fair point that this study couldn’t influence the conception and design of the current study (and therefore couldn’t be included in the current study). However, it is standard practice is to review relevant literature – at the time of manuscript writing – and include new papers to highlight “what is known”. A publication does not necessarily dismiss the need for another study (or a screening tool in this case). However, the authors should appreciate the literature published at the time of the writing and highlight what gap this current paper adds to what is known. The authors seem to have ignored the advancement in the assessment of neuropathic pain in Nepal (i.e., cross-cultural adaptation of the validation of the SLANSS, https://pubmed.ncbi.nlm.nih.gov/34583020/) in the Introduction and Discussion of the current paper. As I see it from my perspective the validation of DN4 in Nepali is an important step in the assessment of neuropathic pain in clinical populations whereas the SLASS was validated in a community sample with the intent for self-administration in community samples. These two measures complement each other and perhaps there is a nice opportunity to compare these two instruments in future research. For tips around writing clear Introduction in the field of Epidemiology, you may refer to https://pubmed.ncbi.nlm.nih.gov/23497856/.

2. Methods: Line 189, specify what criteria were considered for a probable diagnosis of neuropathic pain. Specify what patient’s history and clinical examination contributed to this decision. As indicated in the previous review, this is critical for the internal validity of the study.

3. Line 208, I would have thought ICC agreement is suitable rather than “consistency”. Justify the relevance of “consistency” over “agreement”. Similarly, why is it ICC(3,1) and not ICC(2,1)?

4. Results: Table 4. A follow-up clarification to the previous comment. Authors state that in their response to reviewers that “Table 4 shows the ICC of internal consistency reliability”. However, this ICC is used for test-retest reliability NOT for internal consistency. Cronbach’s alpha is used for internal consistency unless there is other credible evidence for this. Please add such references to justify your choices.

5. All results related to ICC and Cronbach’s alpha should be reported in decimals (report up to 3 digits after decimals).

6. The Discussion section still remains suboptimal (also corroborated by Reviewer 3). The authors also ignored the previous comments to adopt an established approach to reporting the Discussion section. In order to meet the requirements for the journal, please adopt PLOS’s recommendation on writing Discussion (see https://plos.org/resource/how-to-write-conclusions/). Lines 291-302 (of the track changed version of the manuscript) are not relevant to the Discussion section and should be deleted. Sections from lines 303 to 319 may be moved to Results but should be abbreviated by retaining “Results” aspects of the texts. This section needs restructuring and writing.

7. Limitations: authors seem to have ignored two of the previous comments related to the study limitations (also identified by Reviewer 3. First, the authors (all reasons considered), did not assess construct validity using hypothesis testing (see COSMIN guidance on its importance). This should be acknowledged as a limitation and could be promoted as a recommendation for future studies on DN4.

8. The authors did not assess education as their study variable but stated that both literate and illiterate patients were included in the study (lines 352-353). We cannot comment on something that wasn’t measured. Also in the previous response, the authors stated “Our study findings can be generalized to all those who can communicate in Nepali, irrespective of their demographics like education, ethnicity, occupation and income. We also mentioned in the discussion that we did not face any difficulty during data collection because of the difference in literacy level.” This is not true. If this was true, Nepali translation of DN4 wouldn’t be required provided it is shown to be valid in multiple other languages and socioeconomic groups. The study is only generalisable to the population on which they are tested. If the sample consisted of all participants with high levels of education, the study can’t be generalised to those who can’t read or write. Social determinants of health are important in research and health care, also endorsed by the World Health Organisation (https://www.who.int/health-topics/social-determinants-of-health#tab=tab_1). See more on social determinants of health here (https://pubmed.ncbi.nlm.nih.gov/24189091/ and https://www.ncbi.nlm.nih.gov/pmc/articles/PMC3863696/). Not assessing socioeconomic factors is an important limitation and future studies should consider assessing these.

Reviewers' comments:

Reviewer's Responses to Questions

**Comments to the Author**

1. If the authors have adequately addressed your comments raised in a previous round of review and you feel that this manuscript is now acceptable for publication, you may indicate that here to bypass the “Comments to the Author” section, enter your conflict of interest statement in the “Confidential to Editor” section, and submit your "Accept" recommendation.

Reviewer #3: (No Response)

2. Is the manuscript technically sound, and do the data support the conclusions?

Reviewer #3: Yes

3. Has the statistical analysis been performed appropriately and rigorously? 

Reviewer #3: Yes

4. Have the authors made all data underlying the findings in their manuscript fully available?

Reviewer #3: Yes

5. Is the manuscript presented in an intelligible fashion and written in standard English?

Reviewer #3: Yes

6. Review Comments to the Author

Reviewer #3: It is my pleasure to review this manuscript that aimed to translate DN4 questionnaire into Nepalese and assess its psychometric properties. The manuscript is clear and well-written. The reviewer appreciates the excellent work of the authors. However, I have few comments for the authors.

Abstract:

“This study aimed to translate and validate DN4 questionnaire into Nepalese version.”

The use of this phrase implies that “validate” is something you do to a test and that “validity” is a property of the test itself.

“Tests do not have reliabilities and validities, only test responses do. This is an important point because test responses are a function not only of the items, tasks, or stimulus conditions but of the persons responding and the context of measurement.” SEE: Messick S. Validity. In: Linn RL, editor. Educational Measurement. 3rd ed. Phoenix: ORYZ Press; 1993. p. 14.

Validity processes are aimed not at the integrity of the measures themselves, but about inferences that can be made about the people who complete those measures.

I would suggest that the term “validate the measure (DN4 questionnaire)” be changed throughout the entire paper. For example, you may recast as: “This study aimed to translate the DN4 questionnaire into Nepalese and assess its psychometric properties.”

Method:

1. Did you obtain permission to translate the DN4 questionnaire?

2. How did the authors handled missing data?

3. As far as the methodology is concern, no any form of validity test (i.e. content, structural and criterion validity) has been conducted even though the authors used the COSMIN checklist.

With the study sample size, the authors could have at least assess structural validity of the Nepalese DN4 questionnaire

4. “The Area under the Curve (AUC) was considered as the area between the two episodes of the tests” What do you mean by two episodes? Are you referring to test and retest? If so, it sounds more appropriate to use the latter.

5.

Discussion:

I would suggest to the authors to rearrange the flow of the discussion as follows:

a. Translation and cultural adaptation

b. General characteristics including response of the questionnaire

c. Responsiveness (sensitivity to change)

d. Reliability

e. Strengths and limitations

f. Implication to clinical practice and research

Good luck

7. PLOS authors have the option to publish the peer review history of their article (what does this mean?). If published, this will include your full peer review and any attached files.

Reviewer #3: **Yes: **Dr. Aminu A Ibrahim

---

## [Author Response · Author response to Decision Letter 1]

3 Jan 2023

To

Edrian Nim Tolentino

PLOS ONE

RE: PONE-D-21-36866R2 - “Nepalese version of Douleur Neuropathique 4 (DN4) questionnaire for detection of neuropathic pain: translation and assessment of psychometric properties”

Dear Edrian, 

Thank you so much for your email received on 2nd January 2023 requesting the edits. We are very grateful that you highlighted the error in our manuscript. We have now removed the ethics statement which was repeated besides in the methods section. 

Please find attached the following for your kind consideration:

1) A marked-up copy of the manuscript highlighting changes as per the editor’s and reviewers’ comments. 

2) A clean version of the revised manuscript without track changes

Hope the changes have been made as per your suggestion. We are always available for any further information if required. 

Yours Sincerely,

Sujata Shakya

Corresponding Author

---

## [Decision Letter · Decision Letter 2]

23 Feb 2023

PONE-D-21-36866R2Nepalese version of Douleur Neuropathique 4 (DN4) questionnaire for detection of neuropathic pain: translation and assessment of psychometric propertiesPLOS ONE

Dear Dr. Shakya,

Thank you for submitting your manuscript to PLOS ONE. After careful consideration, we feel that it has merit but does not fully meet PLOS ONE’s publication criteria as it currently stands. Therefore, we invite you to submit a revised version of the manuscript that addresses the points raised during the review process.

We look forward to receiving your revised manuscript.

Kind regards,

Saurab Sharma, Ph.D.

Guest Editor

PLOS ONE

Additional Editor Comments (if provided):

Dear authors,

Thank you for making extensive revision on the paper and (partly) addressing reviewer’s and my comments. While some aspects of the manuscript have improved, some feedback remain unaddressed and new problems have been identified.

During the resubmission, please check all files are uploaded and correct. These errors only delay the process and do not help any parties involved. Inability to address the comments adequately will lead to rejection of the paper as several issues remain, statistical analyses are questionnaire and manuscript writing quality is still suboptimal for PLOS One.

Please read the feedback carefully and address them adequately. Take enough time to consult statisticians to correct the analyses and reporting of the analyses. Please have a native English speaker review the manuscript for grammar, flow and coherence. The manuscript is still hard to read and interpret even after 2nd round of revision.

When you make changes, refer to page numbers and line numbers in the track changed version. If there are any questions, please feel free to write to me or organise a call to discuss the feedback. I want to be helpful.

Methods:

1. Report completely the scoring of DN4 (that is 0 to 10) and “presence of neuropathic pain signs and symptoms” and “absence of neuropathic signs and symptoms” using a cut score of 4 if this is true. You also write later that cut off 4 yielded the highest sensitivity and specificity, I am unsure where that interpretation came from.

2. Explicitly say, which of these scorings were used for specific data analyses. Example, dichotomous scoring of “yes” or “no” for neuropathic pain for Kappa and 0 to 10 score for ICC (note both of these are reliability test).

3. Be specific with your statements. You write “Under descriptive statistics, mean, frequency and percentage were calculated.” Mean for which outcome or variable? Frequency for which? Etc….

4. Cohen’s Kappa is used for test-retest reliability for dichotomous outcomes. You seem to have used it for diagnostic accuracy. See your text, “To compare the agreement between the score of DN4 questionnaire and the physician’s diagnosis of pain, Cohen’s kappa statistics was used”. This is incorrect but you are welcome to provide a rebuttal. If so, cite a credible reference to support this.

5. In response to my editor’s comment in the previous round, you declined the suggestion of ICC2,1 (2 way random model) and retained the use of ICC3,1 (2 way mixed model) without any convincing explanation. Your choice of ICC3,1 means that your analyses will not be generalisable to other clinicians using DN4 but is only valid for the clinicians in the study. This contradicts with your conclusion that “…it can now be used as a screening tool for ….” For more reading https://www.ncbi.nlm.nih.gov/pmc/articles/PMC4913118/.

6. ROC: You say “We then derived Receiver Operating Characteristic (ROC) Curve for analyzing the discriminative ability of the test. The ROC curve was plotted for two episodes of the test, with sensitivity versus 1-specificity over all possible cutpoints.” Discriminatory ability of which test and against which test? Could you clarify what you mean by “two episodes of the test”.

7. What type of scoring of DN4 was used for ROC curve (dichotomous versus 0-10)?

8. You randomly introduce the use of Responsiveness as a title for a paragraph in Discussion. The study did not aim to assess responsiveness and is not applicable to this study.

9. “Similarly, different diagnostic values to detect neuropathic pain like sensitivity, specificity, positive and negative predictive value, and positive and negative likelihood ratio were calculated.” – Say how?

10. Use of 2x2 contingency table would be useful for the readers with diagnosis of neuropathic pain signs and symptoms by the index test (using DN4’s dichotomous categories) and reference test (doctor’s diagnosis).

11. You use PPV and NPV without spelling them out for the first time in the main text.

Results:

12. Please upload PDF version of the Nepali DN4 instead of TIFF version. Make sure to refer ALL the Supplementary Tables and Figures in the Main text.

13. Table 2 reports comparison of frequency of items of DN4. But statistical tests used to for this is not presented in Methods section (name of the test in table legend does not suffice). Nor this was an aim of the study. Please list the complete aims of the study. In the same order, present the data analysis methods. Likewise, use the same order to present your results and discuss it. This logical order makes it easier for the readers to read and understand the findings. It is also easier for authors to write in this structure.

14. Tables 2 and 3 uses the diagnosis of pain as “nociceptive” for participants who were not categorised as having “neuropathic”. Not-being neuropathic does not automatically qualify as “nociceptive”. I recommend authors to use “non-neuropathic” instead. Please make sure to use this consistently and throughout the manuscript (like you use in Table 5).

15. Delete the use of “reliability” after “internal consistency” throughout the paper.

16. Line 348: use “Diagnostic accuracy of DN4” instead of ROC curve analysis as the heading.

17. Table 4, use the words and phrases exactly as they appear in the original DN4 questionnaire for ALL items. Eg Painful cold instead of cold pain, Pins and needles instead of Needle sensation. These may mean different things.

18. On line 353, Fig 1 is referred but it isn’t located in the main text. All figures referred to as Figures should be presented in the main text rather than in online supplement.

19. When you say the best sensitivity and specificity scores were identified for a cut off of 4. How was this determined? There is no information on the manuscript around this.

20. Line 355, “Similarly, other measures like Youden index, positive and negative predictive values were also calculated (Table 6).” This info is relevant for Methods section with more details on how to interpret them. There is no mention of Youden index in data analyses section.

Discussion:

21. In paragraph 1, summarise the key results instead of just listing the aims and data analysis plan.

22. “DN4 questionnaire was first created by French neuropathic team and was originally in French language. Due to its simplistic nature, the questionnaire has been translated into various other languages. However, translation can be a complicated process. The major hurdle is the fact that every language has its own unique nuances, grammatical rules, vocabulary, and so on. Rendering ideas from one language to another while also keeping the essence intact is a difficult task with a simple translation” is background information and is not relevant for Discussion. Please remove it. Instead only discuss the results of the current study.

23. Line 399: “In the Hindi version of DN4 questionnaire also, cold pain was present in only 14% of patients with neuropathic pain [9]. In our study, only 17 % of patients with neuropathic pain had symptoms of cold pain.” This info does not fit under the heading of “translation…”. Only include issues related to translation here.

24. In this paragraph, you mention about “cold pain” only. How about other items on the scale? It was help readers with this info for their clinical practice. In our own previous study to identify how people describe their chronic pain, we found a lot of interesting words and phrases to describe pain by Nepali people with chronic pain (https://pubmed.ncbi.nlm.nih.gov/27895511/). If there were no problems with other items, please say so.

25. Line 439: as above, unsure how you concluded that 4/10 was the best cut off point. In fact this wasn’t the main your aim nor you tested this. I recommend you delete this throughout.

26. As above, replace responsiveness heading (line 441) with Diagnostic accuracy. Also, Kappa isn’t a measure for diagnostic accuracy but is for reliability. Your analyses should be revised and correct both for reliability and diagnostic accuracy.

27. Line 465 and 466, in the previous round of review, I pointed out that ICC values should be in decimal places, and your response was that this was corrected in all locations although it was’t. As an editor, it is frustrating when authors say the errors are corrected when they are not. It can be perceived as attempt to deceive.

28. Strengths and limitations: unsure how interviews minimize reliability and validity? Interviews in fact can lead to socially desirable response and interviewers can influence the responses affecting the validity of the responses. Also, DN4 has components that need clinical assessment unlike self-reported measures. So, this discussion point is irrelevant, and I recommend deleting.

29. Limitation 1: “The patients were first seen by a pain physician who was not involved in the interview.” This isn’t a limitation really but is in fact a strength. Ideally, the index test and the reference test should be conducted independently maintaining blinding by separate clinicians. This blinding ensures the robustness of the diagnostic accuracy tests. See QUADAS-2 tool for more details (https://pubmed.ncbi.nlm.nih.gov/22007046/). I also recommend using the terms such as index test and reference test to describe the tests as subheadings in Methods section.

30. The limitation here should be, in line with my comments in the previous round of review, “reference test for diagnosis of neuropathic signs and symptoms was performed by a pain physician using clinical examination alone” or something along this line.

31. Implication: while it is true that DN4 could be used in epidemiological studies, but it will demand enormous resources as it will require trained clinicians to perform physical examination increasing the costs of the study significantly as opposed to using self-reported tools alone. This further contradicts with authors’ point of its utility in “developing countries” which are resource limited. I would recommend authors to align the discussion around this or focus the implications around the clinical utility of DN4 rather than on epidemiological studies. Relevant to this, please change the use of “epidemiological” from conclusions of your abstract to “research settings”

Other comments:

32. A general comment: as DN4 is a screening or diagnostic tool, test-retest reliability isn’t the primary concern here. Instead the focus should have been on inter- or intra-rater reliability. Test-retest reliability is critical for tests that are prognostic or that are used as outcome measures to track change within group or difference between groups.

33. Title: I recommend adding detection of neuropathic pain “signs and symptoms” in order to factor in its limitation of diagnostic accuracy in line with my feedback in the previous round of the review.

34. Abstract: Needs to be significantly revised in order to give it a structure that I was referring in above. Aims should include all aspects of measurement properties: internal consistency, test-retest reliability, diagnostic accuracy. In Methods, write how these were assessed in the same order. Results: report the results also in the exact same order. Unclear what “strength of test” means in line 45. It is confusing to read about AUC without any mention in Methods. Also, there was no mention of first and second test in the Methods.

Good luck.

Reviewers' comments:

Reviewer's Responses to Questions

**Comments to the Author**

1. If the authors have adequately addressed your comments raised in a previous round of review and you feel that this manuscript is now acceptable for publication, you may indicate that here to bypass the “Comments to the Author” section, enter your conflict of interest statement in the “Confidential to Editor” section, and submit your "Accept" recommendation.

Reviewer #3: All comments have been addressed

2. Is the manuscript technically sound, and do the data support the conclusions?

Reviewer #3: Yes

3. Has the statistical analysis been performed appropriately and rigorously? 

Reviewer #3: Yes

4. Have the authors made all data underlying the findings in their manuscript fully available?

Reviewer #3: Yes

5. Is the manuscript presented in an intelligible fashion and written in standard English?

Reviewer #3: Yes

6. Review Comments to the Author

Reviewer #3: I have no further comments as the authors have addressed all my previous comments adequately. I congratulate the authors for the excellent efforts.

7. PLOS authors have the option to publish the peer review history of their article (what does this mean?). If published, this will include your full peer review and any attached files.

Reviewer #3: **Yes: **Dr. Aminu A. Ibrahim

---

## [Author Response · Author response to Decision Letter 2]

6 Apr 2023

To

Dr. Saurab Sharma

Guest Editor, PLOS ONE

RE: PONE-D-21-36866R2 - “Nepalese version of Douleur Neuropathique 4 (DN4) questionnaire for detection of neuropathic pain signs and symptoms: translation and assessment of psychometric properties”

Dear Dr. Saurab Sharma, 

Thank you so much for your email received on 23rd February 2023 and the valuable suggestions. We have attempted our best to address all the comments and give a point-by-point responses.

Please find attached the following for your kind consideration:

1) A rebuttal letter that responds to each comment of the editor

2) A marked-up copy of the manuscript highlighting changes as per the editor’s comments. 

3) A clean version of the revised manuscript without track changes

Hope the changes made have addressed all your comments and suggestions. We are always available for any further information if required. 

Yours Sincerely,

Sujata Shakya

Corresponding Author

POINT BY POINT RESPONSES TO THE COMMENTS OF THE EDITOR

PONE-D-21-36866R2

Response to the editor 

• Response: We thank the editor for reviewing the manuscript and very useful and encouraging comments. We have revised the manuscript in line with your suggestions. We have also done language corrections by consulting with the English language expert. The specific changes and responses to the different points raised are mentioned below. We have specified the respective line numbers based on the track changed version of the revised manuscript. Your comments are highlighted in bold followed by our responses.

Methods:

1. Report completely the scoring of DN4 (that is 0 to 10) and “presence of neuropathic pain signs and symptoms” and “absence of neuropathic signs and symptoms” using a cut score of 4 if this is true. You also write later that cut off 4 yielded the highest sensitivity and specificity, I am unsure where that interpretation came from.

• Response: Thank you for your suggestions. In our revised manuscript track change version line number 244-246, we explained about using cut-off score of 4 for dichotomous categories of Pain assessment findings. We have now deleted ROC curve and all interpretations on this based on the earlier online video conversation. We also removed the words ‘cut-off 4 yielded the highest sensitivity and specificity’ of line numbers 395, 396, 465, 466.

2. Explicitly say, which of these scorings were used for specific data analyses. Example, dichotomous scoring of “yes” or “no” for neuropathic pain for Kappa and 0 to 10 score for ICC (note both of these are reliability test).

• Response: We have now clearly indicated the scoring that we used for ICC and kappa in line numbers 274, 275 of track changed version. We have included the kappa coefficient for assessing reliability test in table 6 and removed from table 3.

3. Be specific with your statements. You write “Under descriptive statistics, mean, frequency and percentage were calculated.” Mean for which outcome or variable? Frequency for which? Etc…. 

• Response: Thank you for your suggestions. We have revised and now mentioned the specific descriptive statistics for the respective variables in line numbers 238-241 of track changed version.

4. Cohen’s Kappa is used for test-retest reliability for dichotomous outcomes. You seem to have used it for diagnostic accuracy. See your text, “To compare the agreement between the score of DN4 questionnaire and the physician’s diagnosis of pain, Cohen’s kappa statistics was used”. This is incorrect but you are welcome to provide a rebuttal. If so, cite a credible reference to support this. 

• Response: Thank you so much for your comments. We now realized that we made a mistake by using Cohen’s kappa for diagnostic accuracy. It should be used for test retest reliability. Now we have included Cohen’s kappa statistics for primarily calculating test retest reliability (table 6) and then ICC as the secondary analysis (S3_Table). We have explained this in our methods section in line numbers 274, 275.

5. In response to my editor’s comment in the previous round, you declined the suggestion of ICC2,1 (2 way random model) and retained the use of ICC3,1 (2 way mixed model) without any convincing explanation. Your choice of ICC3,1 means that your analyses will not be generalisable to other clinicians using DN4 but is only valid for the clinicians in the study. This contradicts with your conclusion that “…it can now be used as a screening tool for ….” For more reading https://www.ncbi.nlm.nih.gov/pmc/articles/PMC4913118/.

• Response: We have now changed ICC from ICC(3,1) to ICC(2,1) in line numbers 279-280 since we agreed that using ICC(2,1) indicates generalizability of the study findings to other clinicians and researchers.

6. ROC: You say “We then derived Receiver Operating Characteristic (ROC) Curve for analyzing the discriminative ability of the test. The ROC curve was plotted for two episodes of the test, with sensitivity versus 1-specificity over all possible cutpoints.” Discriminatory ability of which test and against which test? Could you clarify what you mean by “two episodes of the test”. 

• Response: As per the suggestion during online conversation, we removed ROC curve from our data analysis and results and only included the 2X2 contingency table to compare the index test and reference test (table 3).

7. What type of scoring of DN4 was used for ROC curve (dichotomous versus 0-10)? 

• Response: We have now deleted the ROC curve from our data analysis.

8. You randomly introduce the use of Responsiveness as a title for a paragraph in Discussion. The study did not aim to assess responsiveness and is not applicable to this study. 

• Response: We are really grateful to the editor for pointing out this error. We removed the word ‘responsiveness’ from the discussion section and changed it to diagnostic accuracy in line number 455.

9. “Similarly, different diagnostic values to detect neuropathic pain like sensitivity, specificity, positive and negative predictive value, and positive and negative likelihood ratio were calculated.” – Say how? 

• Response: Thank you for your suggestions. In the revised version, we described specifically in the methods section how each of the above measures were calculated in our study. These were explained in line numbers 249-265.

10. Use of 2x2 contingency table would be useful for the readers with diagnosis of neuropathic pain signs and symptoms by the index test (using DN4’s dichotomous categories) and reference test (doctor’s diagnosis). 

• Response: We included 2X2 contingency table highlighting the association between the diagnosis of neuropathic pain signs and symptoms by the index test and the reference test (table 3).

11. You use PPV and NPV without spelling them out for the first time in the main text. 

• Response: We thank the editor for this comment. We have now done corrections accordingly in line numbers 257-259.

Results:

12. Please upload PDF version of the Nepali DN4 instead of TIFF version. Make sure to refer ALL the Supplementary Tables and Figures in the Main text .

• Response: We have uploaded the PDF version of the Nepali DN4 tool as suggested. We also included short description of the supplementary tables and figures in the main text of the manuscript.

13. Table 2 reports comparison of frequency of items of DN4. But statistical tests used to for this is not presented in Methods section (name of the test in table legend does not suffice). Nor this was an aim of the study. Please list the complete aims of the study. In the same order, present the data analysis methods. Likewise, use the same order to present your results and discuss it. This logical order makes it easier for the readers to read and understand the findings. It is also easier for authors to write in this structure. 

• Response: Thank you so much for your suggestion. We have removed the statistical test in table 2 and now presented only the frequency table of different items of DN4 questionnaire in neuropathic and non-neuropathic pain signs and symptoms (table 2). We have included this in our methods section as well in line numbers 240, 241. As suggested, we have tried our best to include all the explanations in each section of the manuscript in the same logical order.

14. Tables 2 and 3 uses the diagnosis of pain as “nociceptive” for participants who were not categorised as having “neuropathic”. Not-being neuropathic does not automatically qualify as “nociceptive”. I recommend authors to use “non-neuropathic” instead. Please make sure to use this consistently and throughout the manuscript (like you use in Table 5). 

• Response: We totally agree with the editor’s comments and now changed the word from ‘nociceptive’ to ‘non-neuropathic’ in tables 2 and 3 and throughout the manuscript.

15. Delete the use of “reliability” after “internal consistency” throughout the paper. 

• Response: As per the suggestion, we have deleted the word ‘reliability’ throughout and just included the words ‘internal consistency’.

16. Line 348: use “Diagnostic accuracy of DN4” instead of ROC curve analysis as the heading. 

• Response: We have now deleted the ROC curve from our analysis.

17. Table 4, use the words and phrases exactly as they appear in the original DN4 questionnaire for ALL items. Eg Painful cold instead of cold pain, Pins and needles instead of Needle sensation. These may mean different things. 

• Response: Thank you for your valuable comments. We totally agree that the words should exactly be same as in the original questionnaire. We have kept the words and phrases that were used in the original DN4 questionnaire throughout the manuscript and also in table 2 and table 5.

18. On line 353, Fig 1 is referred but it isn’t located in the main text. All figures referred to as Figures should be presented in the main text rather than in online supplement. 

• Response: We have removed figure 1 and now made sure that each table is presented in the main text and not in the online supplement.

19. When you say the best sensitivity and specificity scores were identified for a cut off of 4. How was this determined? There is no information on the manuscript around this. 

• Response: We apologize for this error. Now we have deleted the sentences of line numbers 395, 396, 465, 466.

20. Line 355, “Similarly, other measures like Youden index, positive and negative predictive values were also calculated (Table 6).” This info is relevant for Methods section with more details on how to interpret them. There is no mention of Youden index in data analyses section. 

• Response: Thank you for your comments. We have now described the use of Youden index and its explanation in the methods section in line numbers 254-257.

Discussion:

21. In paragraph 1, summarise the key results instead of just listing the aims and data analysis plan. 

• Response: The summary of key results has been added in line numbers 418-423.

22. “DN4 questionnaire was first created by French neuropathic team and was originally in French language. Due to its simplistic nature, the questionnaire has been translated into various other languages. However, translation can be a complicated process. The major hurdle is the fact that every language has its own unique nuances, grammatical rules, vocabulary, and so on. Rendering ideas from one language to another while also keeping the essence intact is a difficult task with a simple translation” is background information and is not relevant for Discussion. Please remove it. Instead only discuss the results of the current study. 

• Response: The whole paragraph has been removed of line numbers 427-432.

23. Line 399: “In the Hindi version of DN4 questionnaire also, cold pain was present in only 14% of patients with neuropathic pain [9]. In our study, only 17 % of patients with neuropathic pain had symptoms of cold pain.” This info does not fit under the heading of “translation…”. Only include issues related to translation here. 

• Response: This part has been removed from heading of translation of line numbers 442-444.

24. In this paragraph, you mention about “cold pain” only. How about other items on the scale? It was help readers with this info for their clinical practice. In our own previous study to identify how people describe their chronic pain, we found a lot of interesting words and phrases to describe pain by Nepali people with chronic pain (https://pubmed.ncbi.nlm.nih.gov/27895511/). If there were no problems with other items, please say so. 

• Response: We have added discussion about the different words use by Nepalese to describe electric pain in line numbers 432-438. Earlier we did not include this in the discussion but later we realize that the readers will also know about different words to describe the same item. Thank you for suggesting.

25. Line 439: as above, unsure how you concluded that 4/10 was the best cut off point. In fact this wasn’t the main your aim nor you tested this. I recommend you delete this throughout. 

• Response: We have now deleted that 4/10 is best cut off point in our study of line numbers 453, 454.

26. As above, replace responsiveness heading (line 441) with Diagnostic accuracy. Also, Kappa isn’t a measure for diagnostic accuracy but is for reliability. Your analyses should be revised and correct both for reliability and diagnostic accuracy. 

• Response: We have revised this mistake and now replaced the heading responsiveness with diagnostic accuracy in line number 455. We included in the discussion section under the heading reliability that kappa was used for test-retest reliability in line number 478.

27. Line 465 and 466, in the previous round of review, I pointed out that ICC values should be in decimal places, and your response was that this was corrected in all locations although it was’t. As an editor, it is frustrating when authors say the errors are corrected when they are not. It can be perceived as attempt to deceive. 

• Response: We are really sorry for this mistake. We failed to correct this mistake in discussion part in our earlier revision. Now, it has been corrected in line numbers 477, 478. 

28. Strengths and limitations: unsure how interviews minimize reliability and validity? Interviews in fact can lead to socially desirable response and interviewers can influence the responses affecting the validity of the responses. Also, DN4 has components that need clinical assessment unlike self-reported measures. So, this discussion point is irrelevant, and I recommend deleting. 

• Response: This line has been deleted (line numbers 486-488).

29. Limitation 1: “The patients were first seen by a pain physician who was not involved in the interview.” This isn’t a limitation really but is in fact a strength. Ideally, the index test and the reference test should be conducted independently maintaining blinding by separate clinicians. This blinding ensures the robustness of the diagnostic accuracy tests. See QUADAS-2 tool for more details √ (https://pubmed.ncbi.nlm.nih.gov/22007046/). I also recommend using the terms such as index test and reference test to describe the tests as subheadings in Methods section. 

• Response: We have included this under strength in line numbers 488, 490. We have used index test and reference test in Methods section (line numbers 219-221, 227, 230) and throughout the manuscript.

30. The limitation here should be, in line with my comments in the previous round of review, “reference test for diagnosis of neuropathic signs and symptoms was performed by a pain physician using clinical examination alone” or something along this line. 

• Response: We highlighted that our study limitation was that the reference test for diagnosis was based on history taking and clinical examination (line numbers 492-497). 

31. Implication: while it is true that DN4 could be used in epidemiological studies, but it will demand enormous resources as it will require trained clinicians to perform physical examination increasing the costs of the study significantly as opposed to using self-reported tools alone. This further contradicts with authors’ point of its utility in “developing countries” which are resource limited. I would recommend authors to align the discussion around this or focus the implications around the clinical utility of DN4 rather than on epidemiological studies. Relevant to this, please change the use of “epidemiological” from conclusions of your abstract to “research settings” 

• Response: The DN4 questionnaire required physical examination which can be performed only by trained medical personnel, so we realized that this DN4 questionnaire is relevant in research setting so the word epidemiological has been removed throughout (line numbers 69, 103, 108, 509) and the implication has been highlighted towards the clinical utility of the questionnaire in line numbers 511, 512. Thanks for this analysis.

Other comments:

32. A general comment: as DN4 is a screening or diagnostic tool, test-retest reliability isn’t the primary concern here. Instead the focus should have been on inter- or intra-rater reliability. Test-retest reliability is critical for tests that are prognostic or that are used as outcome measures to track change within group or difference between groups.

• Response: We admit that the best approach would have been using inter-rater reliability. However, we could not perform it due to difficult clinical settings of our hospital. 

33. Title: I recommend adding detection of neuropathic pain “signs and symptoms” in order to factor in its limitation of diagnostic accuracy in line with my feedback in the previous round of the review. 

• Response: The neuropathic sign and symptoms has been added throughout the manuscript.

34. Abstract: Needs to be significantly revised in order to give it a structure that I was referring in above. Aims should include all aspects of measurement properties: internal consistency, test-retest reliability, diagnostic accuracy. In Methods, write how these were assessed in the same order. Results: report the results also in the exact same order. Unclear what “strength of test” means in line 45. It is confusing to read about AUC without any mention in Methods. Also, there was no mention of first and second test in the Methods. 

• Response: The study aim in the abstract has been revised accordingly (line numbers 32, 33) and we now revised the order of reporting the measures and results in all sections of the abstract.

We are really grateful to the editor for helping us take the manuscript to this stage. We hope that the amendments made would meet your expectations. We are completely open to any further suggestions.

Your's sincerely

Sujata Shakya

Corresponding author

---

## [Editor Report · Decision Letter 3]

13 Jun 2023

Nepalese version of Douleur Neuropathique 4 (DN4) questionnaire for detection of neuropathic pain signs and symptoms: translation and psychometric properties

PONE-D-21-36866R3

Dear Dr. Shakya,

We’re pleased to inform you that your manuscript has been judged scientifically suitable for publication and will be formally accepted for publication once it meets all outstanding technical requirements.

Kind regards,

Saurab Sharma, Ph.D.

Guest Editor

PLOS ONE

Additional Editor Comments (optional):

Please note that several typographical errors still exist in the manuscript which I believe can be fixed at the typesetting and proofreading stages. 
---

## [Editor Report · Acceptance letter]

7 Jul 2023

PONE-D-21-36866R3 

Nepalese version of Douleur Neuropathique 4 (DN4) questionnaire for detection of neuropathic pain signs and symptoms: translation and psychometric properties 

Dear Dr. Shakya:

I'm pleased to inform you that your manuscript has been deemed suitable for publication in PLOS ONE. Congratulations! Your manuscript is now with our production department. 

Kind regards, 

on behalf of

Dr. Saurab Sharma 

Guest Editor

PLOS ONE